# Levels of Burnout and Engagement after COVID-19 among Psychology and Nursing Students in Spain: A Cohort Study

**DOI:** 10.3390/ijerph20010377

**Published:** 2022-12-26

**Authors:** Raimundo Aguayo-Estremera, Gustavo R. Cañadas, Elena Ortega-Campos, Laura Pradas-Hernández, Begoña Martos-Cabrera, Almudena Velando-Soriano, Emilia I. de la Fuente-Solana

**Affiliations:** 1Departamento de Psicobiología y Metodología de las Ciencias del Comportamiento, Facultad de Psicología, Campus de Somosaguas, Ctra. De Húmera, s/n, Universidad Complutense de Madrid, Pozuelo de Alarcón, 28223 Madrid, Spain; 2Department of Didactic of Mathematics, Faculty of Education Science, Campus de Cartuja s/n, University of Granada, 18071 Granada, Spain; 3Centro de Investigación en Salud-UAL (CEINSA-UAL), Carretera de Sacramento, s/n, La Cañada de San Urbano, Universidad de Almería, 04120 Almería, Spain; 4San Cecilio Clinical University Hospital, Avenida del Conocimiento, s/n, Andalusian Health Service, 18016 Granada, Spain; 5Brain, Mind and Behaviour Research Center (CIMCYC), Campus de Cartuja s/n, University of Granada, 18071 Granada, Spain

**Keywords:** burnout, COVID-19, engagement, students, cohort study

## Abstract

The COVID pandemic has 0drastically changed the functioning of universities in Spain and may have altered individuals’ behaviours and emotions, the way they engage in the learning process and their psychological well-being. Burnout syndrome is a psychological problem that arises from persistent confrontation with emotional and interpersonal stressors. COVID-related burnout among Spanish students has received little research attention. For this study, a pre-post cohort study design was used. Data were collected using the Maslach Burnout Inventory—Student Survey, the Granada Burnout Questionnaire for university students, the Utrecht Work Engagement Scale and the Fear of CoronaVirus-19 scale. The population was composed of two samples of 190 and 226 students from Spanish universities. According to the results obtained, significant differences were observed between the pre- and post-test samples. Levels of burnout were higher after the COVID-19 pandemic and students’ levels of engagement have dropped significantly following their experiences of the COVID pandemic. This study shows the impact that the covid pandemic has had on Spanish university students, impacts which may have had important consequences for their mental and physical health. It is necessary to implement intervention programs to enable students to recover, at least, the levels of burnout and engagement prevailing before the outbreak of the pandemic.

## 1. Introduction

The COVID-19 pandemic has forced governments in most countries to enact regional and personal lockdowns. In schools, colleges and universities, this situation has disrupted normal teaching dynamics and provoked a sharp increase in the use of online methods. The sudden implementation of this new approach has changed the way in which teachers and students communicate and interact, impacting on fundamental aspects of education, such as the presentation of content and the evaluation of knowledge [1,2].

The rapid transmission of COVID-19 throughout the world has had major consequences for individuals’ physical and psychological health, and in social, economic and educational spheres. In an immediate reaction, schools and colleges closed their doors to prevent the spread of infection. The United Nations Educational, Scientific and Cultural Organisation [3] estimated that, by March 2020, 90% of all students worldwide had been affected by this closure of educational centres. At the beginning of the 2020–2021 school year, over 850 million students were still unable to access their classrooms.

These events also affected Spain. On 14 March 2020, the Spanish government declared a state of emergency, imposing a strict lockdown in which children and teenagers were confined to their homes for over six weeks. A de-escalation phase then began, during which these restrictions were gradually relaxed, until the beginning of July (under the Royal Decree 463/2020, of 14 March, there was declared a nationwide state of alarm in order to manage the health crisis caused by COVID-19 [4]). Thus, schools and universities were forced to switch their teaching, almost overnight, to a virtual mode, which was maintained throughout the 2019–2020 academic year [5]. The subsequent 2020–2021 academic year, too, was marked by improvised reactions, with little margin for reflection and planning. Although each university was authorised to determine its own form of education delivery (distance, blended or face-to-face learning) and the timetable for implementing changes, the government recommended a mixed strategy based on face-to-face and distance learning activities [6]. Moreover, for in-person teaching, several recommendations differentiated the new situation from pre-pandemic circumstances; for example, windows were to be kept open, students had to remain physically distanced and face masks were obligatory.

As a result of these developments, it is reasonable to surmise that, for many students, university life became even more stressful than before [7]. Burnout is described as an emotional response to chronic work stress with three dimensions, emotional exhaustion (EE), understood as a progressive loss of energy; depersonalization (D), expressed as hostility towards the work environment; and feelings of low professional accomplishment (PA) or loss of self-confidence and demotivation. This is a psychological problem that arises from persistent confrontation with emotional and interpersonal stressors [8], which may include students’ experiences during the pandemic and the consequent restrictions on their academic lives. These occurrences may have altered individuals’ behaviours and emotions, the way they engage in the learning process and, ultimately, their psychological well-being. This is especially so for students seeking university degrees in health sciences, who would have been closer to the focus of the problem during the pandemic. Indeed, these outcomes have already been observed [7,9,10,11,12], and constitute a risk factor for academic dropout [9,13].

Although some research has been conducted in this field regarding non-university teachers and students [5,14,15], few studies have addressed the question of Spanish university students in particular. In one of the few papers to do so, Merino-Godoy et al. [10] concluded that emotional exhaustion explained part of the stress experienced by nursing students. In another, de la Fuente et al. [7] reported that the methodological difficulties involved in teaching/learning and the excessive workload produced were associated with increased levels of burnout among students. In view of the paucity of research into COVID-related burnout among Spanish students, this paper examines whether levels of burnout and engagement changed among psychology and nursing students following the outbreak of the COVID-19 pandemic. In addition, we consider the differences produced in these psychological variables, in the context of concerns about the coronavirus, taking into account whether the students had suffered from COVID-19, whether someone they knew had died from it or whether they had suffered psychological and physical consequences derived from the social response to the pandemic.

## 2. Materials and Methods

### 2.1. Participants

Two sample populations were considered, each composed of students from Spanish universities, with 190 and 226 students, respectively. All were studying the first or second year of an undergraduate degree course in psychology or nursing. Recruitment was achieved through non-probabilistic sampling, obtaining a response rate of 100%. The students presented the following characteristics: 74.5% were women, 94.1% were single, 86.3% were not simultaneously in employment, and their ages ranged from 18 to 64 years, with a mean of 21.01 years (SD = 4.67).

### 2.2. Design and Procedure

A pre-post study design with cohorts was used [16,17]. To maximise the similarity between the cohorts, and so reinforce the internal validity of the study, participants with closely matching sociodemographic characteristics were chosen (all were pursuing the same degree courses, with the same syllabus, in the same academic year, with identical access requirements and within the same educational and cultural context). The data were collected in several universities following a standardised procedure in the second semesters (February to April) of 2018 and 2021. The questionnaires were the same for all students in both samples, except for the Fear of Coronavirus-19 scale, which was only administered in the second sample. Data collection took place in the classrooms during teaching time, with the approval of the deans and lecturers involved. The member of the team in charge of administering the questionnaires remained in the classroom with the students for the entire duration of the data collection. The students were asked to participate several days prior to the administration of the questionnaire by a member of the team who came to the classroom to briefly explain the research. During the data collection the teacher of the course was absent from the classroom. The purpose of the first data collection, before the outbreak of COVID-19, was to describe the levels of burnout and engagement in a sample of university students.

### 2.3. Variables and Instruments

An ad hoc sociodemographic questionnaire was used to obtain data on the students’ age, sex, marital status and employment situation. They were also asked questions related to the COVID-19 pandemic: whether they had been infected, whether someone they knew had had the disease or died from it, and whether they had experienced psychological or physical consequences from the social situation arising from the pandemic. In addition, the following measurement instruments were used.

Maslach Burnout Inventory—Student Survey (MBI-SS) [18], with 15 items scored on a seven-point response scale, is used to measure the three dimensions of the syndrome according to the original proposal by Maslach and Jackson [19], namely emotional exhaustion (of psychological and emotional resources), depersonalisation (feelings of cynicism and detachment) and diminished personal accomplishment (feelings of ineffectiveness and inability).

The Granada Burnout Questionnaire for university students (Cuestionario de Burnout Granada—CBG-US) [20] contains 16 items and uses a five-point response scale (ranging from 1, totally disagree, to 5, totally agree) to measure the three dimensions of the syndrome (emotional exhaustion, depersonalisation and personal accomplishment).

The Utrecht Work Engagement scale (UWES) [21] includes 24 items, scored on a seven-point response scale, and measures the three dimensions of engagement: absorption (fully concentrated and happily engrossed in one’s work), dedication (commitment to one’s work and feelings of importance and enthusiasm) and vigour (energy and mental resilience).

The Fear of Coronavirus-19 scale (FCV19-S) [22] consists of seven items scored on a five-point response scale.

### 2.4. Ethics

The study was conducted in accordance with the Declaration of Helsinki and approved by the Institutional Review Board (or Ethics Committee) of the University of Granada (393/CEIH2017). All participants gave prior informed consent and were assured confidentiality and anonymity at all times.

### 2.5. Statistical Methods

Descriptive statistics were calculated for all variables, and equivalence between the two samples was analysed for sociodemographic variables [23]. In this type of analysis, the null hypothesis is that the size of the differences found will be greater than an interval termed the smallest effect size of interest, which was defined as Cohen’s d equal to 0.50 (in absolute terms). Therefore, in order to reject the null hypothesis, we must find evidence of an effect size smaller than the threshold considered to be a difference meaningful.

The effects of COVID-19 on burnout syndrome and engagement among the students were analysed using Welch’s *t*-test [24]. The degree of relationship between the continuous variables was determined by means of a correlation analysis.

Statistical analyses were also performed using the Bayes factor, to test mean-difference hypotheses. Since there was little prior knowledge in this respect, a Cauchy distribution of 0.707 was specified as the prior distribution. A Bayes factor of 1–3 is considered weak evidence for the alternative hypothesis, one of 4–10 is considered moderate evidence, and one greater than 10 is considered strong evidence [25,26]. In addition, credibility intervals were calculated for Cohen’s d.

The reliability of the scales and subscales was estimated from the omega coefficient [27,28]. All statistical analyses were performed using R 4.2.1. [29].

## 3. Results

### 3.1. Sample Equivalence Analysis, Description and Correlation between Variables

Table 1 and Table 2 show the descriptive statistics obtained for all the study variables. The reliability of the scales is demonstrated by the fact that, in all cases and for both samples, the omega coefficients exceeded the commonly recommended value of 0.70. As concerns the variables related to the coronavirus, among the participants in the second sample (N = 226), 47.6% had suffered from COVID-19, 93.8% knew someone close to them who had been infected with the coronavirus, 16.9% had undergone the death of a close person due to the disease, 77.8% believed they suffered psychological consequences from the social situation caused by the pandemic, and 58.4% had suffered physical consequences.

The equivalence analyses revealed that the two samples were equivalent for the following variables: age (t410.4 = −2.48, *p*-value = 0.006), sex (Z = 10.28, *p*-value < 0.001), marital status (Z = −18.99, *p*-value < 0.001) and concurrent employment (Z = 5.77, *p*-value < 0.001).

### 3.2. Burnout, Engagement and Their Relationship with the COVID-19 Pandemic

As shown in Table 3, statistically significant differences were observed between the pre- and post-test samples (i.e., before and after the outbreak of COVID-19) for various dimensions of burnout syndrome and engagement. Specifically, after the outbreak of the pandemic, the students reported lower levels of PA, AB, DE and VI, and greater EE, measured using the CBG-US. Bayesian analysis confirmed these findings. Thus, strong evidence was obtained for several of the hypotheses proposed: after the pandemic, the students had lower PA, AB, DE and VI. In addition, there was moderate evidence for the hypothesis of higher EE in the post-outbreak sample. Strong evidence for the null hypothesis was also observed for D, measured with the CBG-US. The effect sizes (and their 95% credibility intervals) showed that the magnitude of the differences between samples was moderate for burnout syndrome, and moderate to high for engagement.

In the post-pandemic sample, hypothesis tests were conducted to identify differences in the levels of burnout syndrome and engagement according to the coronavirus-related variables considered (see Table 4). Statistically significant results were observed for the variable “psychological consequences”. Moderate to strong evidence was obtained for the hypothesis that students who suffered psychological consequences from the social situation caused by the pandemic presented higher levels of EE and D (measured with the MBI) and lower levels of PA; in addition, they reported greater fear of the coronavirus. Effect sizes ranged from moderate to large. Moderate-to-strong evidence was also found for the hypothesis that those who suffered physical consequences were more afraid of the coronavirus.

The correlations between the study variables, for both samples, were assessed, with the following results (see Table 1 and Table 2). Moderate-to-strong statistically significant correlations were obtained between the dimensions of burnout syndrome, measured with the MBI. The correlation between EE and D was positive, while those between PA and EE, and PA and D were negative. Similar correlations were observed with the CBG-US instrument, although the values obtained between EE and D were smaller. Regarding the dimensions of engagement, all the correlations were statistically significant, strong and positive. Finally, with respect to fear of the coronavirus, statistically significant correlations, positive and of moderate size, were observed for EE and D (measured with the MBI).

## 4. Discussion

In this study, we examined the levels of burnout and engagement experienced before and during the COVID-19 pandemic. To do so, we analysed the survey responses made by two sample populations, one before the outbreak of the pandemic and the other when its effects had become apparent. This type of design, known as a pre-post cohort study [14,15], enables us to analyse, in the manner of a quasi-experimental design, the effect of an independent variable when complete data for the dependent variable are not available. A cohort design can present acceptable internal validity when the cohorts have similar characteristics. For this reason, our study participants had very similar sociodemographic, educational and cultural characteristics: all were first- or second-year undergraduates, studying psychology or nursing. Furthermore, the information collection process was standardised and identical for both samples. Finally, results of the equivalence tests performed on the sociodemographic variables (age, sex, marital status and concurrent employment) indicate that the samples were equivalent (Section 3.2).

According to our study results, after the COVID-19 pandemic the students had higher levels of burnout, especially in terms of greater emotional exhaustion and reduced personal accomplishment (Table 3). These results suggest that the greater impact of stressors related to university studies increased the risk of students’ experiencing burnout syndrome. In this respect, too, the socio-educational consequences of the pandemic (such as isolation, the need for remote or mixed teaching and possible ambiguity in academic assessment systems) were perceived as additional factors for stress [30,31]. According to the job demands–resources theory [32], when an individual believes that the demands of their environment, which for these students is that of university studies, exceed the resources available to address them, the demands are perceived as stressful and can eventually provoke the appearance of burnout and reduce engagement. This theory was borne out in our study, as the students’ levels of engagement were clearly lower following the outbreak of the pandemic, with less absorption, dedication and vigour. This inverse relationship between burnout and engagement is consistent with previous research findings [33].

On the other hand, we found no evidence of higher levels of depersonalisation after COVID-19. Indeed, according to the results obtained with the CBG-US, levels of depersonalisation were lower in the sample that had experienced the events of the pandemic than those obtained three years before. With the MBI, however, no such evidence was found (Table 3). This discrepancy between the CBG-US and the MBI might be explained by the composition of their respective subscales. While the CBG inquires mainly about aspects related to the respondent’s peers (e.g., “I can understand my classmates’ feelings”), the MBI focuses more on organisational issues (e.g., “I doubt the importance and value of my studies”). Moreover, in the situation experienced during the pandemic, it seems reasonable to assume that while emotional exhaustion and the sense of personal fulfilment may worsen, the aspects of depersonalisation will not. Thus, exhaustion and feelings of academic insufficiency may be aggravated by factors such as the new variety of teaching methods (face-to-face and online), ambiguity in the assessment system and physical isolation. However, the experience of the study participants was the same as that of their fellow students, and so they could easily sympathise with them. In accordance with this hypothesis, earlier studies have observed high levels of empathy among students during the pandemic [34,35].

Analysis of the post-outbreak results showed that students who had been infected by COVID-19, or whose friends/relatives had been infected or even died from the disease, did not present increased levels of burnout or reduced levels of engagement (Table 4). This result may be explained by the fact that the latter constructs are associated with the workplace (or academic) environment, unlike others, such as depression and anxiety [36,37], which are more likely to be affected by the above circumstances [38]. Thus, broadly shared events such as exposure to COVID-19 might have only limited impact on personal accomplishment, regarding academic issues, or psychological exhaustion related to daily tasks. A striking result of our analysis is that fear of the coronavirus was not heightened either by personal experience of COVID or by the death of a close acquaintance from this cause, possibly because most of the deaths from the coronavirus affected older adults. In this respect, it has been reported that adults and adolescents differ in their fear of the coronavirus [38].

In the post-outbreak sample population, the students who reported having suffered psychological effects from the social situation due to COVID-19 presented higher levels of emotional exhaustion and depersonalisation (measured with the MBI), and lower levels of personal accomplishment, than those who reported no such effects. This finding shows that the risk of burnout syndrome is one of the psychological consequences that may appear following the social changes caused by COVID-19. Moreover, those who suffered these psychological consequences also reported greater fear of the coronavirus. Emotional exhaustion and fear of the coronavirus were also greater among the students who had had physical consequences from infection (Table 4). These results suggest that what most increases fear of the disease is not so much having experienced it or having close acquaintances who have suffered from it, or even died from it, but rather having suffered psychological or physical consequences from the social changes produced. Finally, fear of illness is associated with increased emotional exhaustion.

In this study, burnout levels were measured with two instruments: on the one hand, the widely used MBI [18,19], and on the other, the CBG-US [20], which was specifically designed for Spanish university students and conforms to the conceptual framework proposed by Maslach and Jackson [19]. The results of previous psychometric studies indicate that the CBG has good convergent validity with the MBI [20,39] and even overcomes one of the usual problems of the MBI, namely the recurrent lack of reliability of the depersonalization dimension [40]. The results obtained in the present study are in line with those previously reported in this regard. In most cases, the results are equivalent for both measurement instruments, both in the pre-pandemic sample and in the post-pandemic sample. Finally, the reliability of the dimensions of the CBG-US was never less than that of the MBI (Table 1 and Table 2).

This study is based on a pre-post cohort design, achieving equivalence of the cohorts by means of design strategies (appropriate criteria for the inclusion of participants) and data analysis (equivalence test and Bayesian analysis). Although these controls might lead us to infer causality and to affirm that all the effects found were due to experiences during the pandemic, the design used does not eliminate all possible threats to the study’s internal validity. For greater confidence before affirming causality, further research in this field, reaching similar conclusions, is necessary.

## 5. Conclusions

High levels of burnout can have serious consequences for the physical and psychological health of university students, including lack of sleep, fatigue, demotivation, poor academic performance, alcohol and drug abuse, absenteeism and the abandonment of studies. Moreover, optimum academic performance requires high levels of engagement, and it is precisely in this area where our results indicate an increased risk of burnout, as the experiences of the COVID-19 pandemic seem to have produced a significant reduction in students’ levels of engagement. For this reason, we recommend that intervention programmes be implemented to enable students to recover, at least, the levels of burnout and engagement prevailing before the outbreak of the pandemic.

### Limitations and Strengths of the Study

This study is based on a pre-post cohort design, achieving equivalence of the cohorts by means of design strategies (appropriate criteria for the inclusion of participants and standardization of data collection) and data analysis (equivalence test and Bayesian analysis). Although these controls might lead us to infer causality and to affirm that all the effects found were due to experiences during the pandemic, the design used does not eliminate all possible threats to the study’s internal validity. This limitation of the present work could be mitigated with further research in this field that reaches similar conclusions. Another limitation of the current study was that results could only be generalized to the population of first- and second-year psychology and nursing undergraduates in Spain. Finally, the lack of statistical power for two of the COVID-19 variables (i.e., “friend or relative infected by” and “friend or relative died from” COVID-19) could explain the obtained results. In this sense, a higher number of participants may reveal differences in burnout and engagement dimensions between students with and without friends or relative infected by or died from COVID-19.

The present paper has several strengths. First, in the study is employed a design that allows for a pre-post COVID-19 comparison on various psychological outcomes, given a rigorous design and statistical procuration to ensure equivalence between groups. Second, the results shed light on how the COVID-19 pandemic situation has affected students’ mental health, specifically in relation to academic tasks. Finally, the conclusions of this study allow a better understanding of how the fear of COVID-19 acts on adolescents, which may change the way in which institutions make a successful media intervention to engage them in healthy behaviours, e.g., vaccination.

It is necessary to develop new research that includes quantitative and qualitative data that provides information on the real impact of the pandemic on health sciences university students across the globe. There is the need for a global analysis of these problems and the approach of models to establish educational policies that can improve the circumstances in which university studies are pursued. 

## Figures and Tables

**Table 1 ijerph-20-00377-t001:** Descriptive statistics and correlations for continuous variables in the first sample.

Variable	N	M	SD	1	2	3	4	5	6	7	8	9	10
1. Age	190	20.8	4.20	-									
2. EE (MBI)	185	13.7	5.87	−0.05	0.78								
3. D (MBI)	183	6.01	5.31	−0.04	0.49 ***	0.79							
4. PA (MBI)	186	27.3	5.86	0.10	−0.18 **	−0.37 ***	0.79						
5. EE (CBG)	187	23.7	5.53	0.10	0.74 ***	0.34 ***	−0.13 *	0.80					
6. D (CBG)	188	13.2	4.03	0.09	0.17 *	0.41 ***	−0.48 ***	0.10	0.80				
7. PA (CBG)	187	27.3	4.22	0.10	−0.39 ***	−0.51 ***	0.59 ***	−0.32 ***	−0.57 ***	0.84			
8. AB	185	27.2	7.23	0.08	−0.15 *	−0.28 ***	0.69 ***	−0.06	−0.40 ***	0.48 ***	0.74		
9. DE	183	40.3	8.15	0.09	−0.24 ***	−0.53 ***	0.78 ***	−0.11 *	−0.56 ***	0.66 ***	0.67 ***	0.92	
10. VI	183	33.3	7.98	0.20 **	−0.16 *	−0.19 **	0.65 ***	−0.08	−0.34 ***	0.53 ***	0.74 ***	0.67 ***	0.74

EE (MBI) = Emotional exhaustion, measured by MBI; D (MBI) = depersonalisation, measured by MBI; PA (MBI) = personal accomplishment, measured by MBI; EE (CBG) = emotional exhaustion, measured by CBG-US; D (CBG) = depersonalisation, measured by CBG-US; PA (CBG) = personal accomplishment, measured by CBG-US; AB = absorption; DE = dedication; VI = vigour. The values on the diagonal are omega coefficients. * *p* < 0.05. ** *p* < 0.01. *** *p* < 0.001.

**Table 2 ijerph-20-00377-t002:** Descriptive statistics and correlations for continuous variables in the second sample.

Variable	N	M	SD	1	2	3	4	5	6	7	8	9	10	11
1. Age	226	19.6	4.63	-										
2. EE (MBI)	225	13.5	6.82	0.13 *	0.86									
3. D (MBI)	222	6.17	5.36	0.27 ***	0.67 ***	0.86								
4. PA (MBI)	224	23.3	6.29	0.04	−0.34 ***	−0.37 ***	0.79							
5. EE (CBG)	221	25.4	7.15	0.25 ***	0.75 ***	0.54 ***	−0.24 ***	0.90						
6. D (CBG)	221	11.9	4.24	0.08	0.17 *	0.31 ***	−0.23 ***	0.18 *	0.85					
7. PA (CBG)	220	24.9	4.91	0.06	−0.63 ***	−0.63 ***	0.59 ***	−0.56 ***	−0.31 ***	0.89				
8. AB	223	21.2	7.07	0.12	−0.28 ***	−0.33 ***	0.71 ***	−0.19 **	−0.16 **	0.54 ***	0.80			
9. DE	222	35.5	9.22	0.03	−0.38 ***	−0.66 ***	0.62 ***	−0.26 ***	−0.40 ***	0.63 ***	0.67 ***	0.92		
10. VI	221	24.9	7.80	0.18 **	−0.43 ***	−0.31 ***	0.60 ***	−0.32 ***	−0.12 *	0.53 ***	0.67 ***	0.49 ***	0.75	
11. FCV	224	14.5	4.98	0.11	0.27 **	0.20 **	0.04	0.22 ***	0.11	−0.02	0.10	−0.04	−0.05	0.82

EE (MBI) = Emotional exhaustion, measured by MBI; D (MBI) = depersonalisation, measured by MBI; PA (MBI) = personal accomplishment, measured by MBI; EE (CBG) = emotional exhaustion, measured by CBG-US; D (CBG) = depersonalisation, measured by CBG-US; PA (CBG) = personal accomplishment, measured by CBG-US; AB = absorption; DE = dedication; VI = vigour. FCV = fear of coronavirus. The values on the diagonal are omega coefficients. * *p* < 0.05. ** *p* < 0.01. *** *p* < 0.001.

**Table 3 ijerph-20-00377-t003:** Effects of the pandemic on burnout and engagement according to the different hypothesis tests.

Variable	*t*	*df*	*p*	*d*	95%CI	BF_10_
EE (MBI)	0.41	407.1	0.683	0.04	−0.15–0.23	0.11
D (MBI)	−0.07	389.7	0.947	0.19	−0.01–0.38	0.67
PA (MBI)	6.89	402.7	<0.001	0.67	0.46–0.86	3.5 × 10^8^
EE (CBG)	−2.68	402.9	0.007	−0.26	−0.45–−0.06	3.43
D (CBG)	3.17	402	0.002	0.31	0.11–0.49	13.391
PA (CBG)	5.19	404.9	<0.001	0.50	0.30–0.69	26,419.76
AB	8.33	389	<0.001	0.82	0.61–1.00	4.4 × 10^12^
DE	5.46	401	<0.001	0.54	0.33–0.72	1.3 × 10^5^
VI	10.60	384.7	<0.001	1.04	0.82–1.24	1.5 × 10^20^

*t* = Student *t*-test; *df* = degrees of freedom; *p* = *p*-value; *d* = standardised mean difference; CI = confidence interval of d; BF_10_ = Bayes factor in favour of the alternative hypothesis; EE (MBI) = emotional exhaustion, measured by MBI; D (MBI) = depersonalisation, measured by MBI; PA (MBI) = personal accomplishment, measured by MBI; EE (CBG) = emotional exhaustion, measured by CBG-US; D (CBG) = depersonalisation, measured by CBG-US; PA (CBG) = personal accomplishment, measured by CBG-US; AB = absorption; DE = dedication; VI = vigour.

**Table 4 ijerph-20-00377-t004:** Results of the hypothesis tests of differences in burnout and engagement according to the coronavirus-related variables, for the second sample.

Variable	Outcome	M (SD)	M (SD)	t (df)	d	95%CI	BF_10_
		No	Yes				
Infected by COVID-19							
	EE (MBI)	13.7 (7.08)	13.3 (6.54)	0.45 (222.9)	0.06	−0.20–0.32	0.16
	D (MBI)	6.2 (5.34)	6.1 (5.41)	0.11 (216.9)	0.01	−0.25–0.28	0.15
	PA (MBI)	23.2 (6.19)	23.4 (6.43)	−0.29 (218.5)	−0.04	−0.29–0.22	0.15
	EE (CBG)	25.4 (7.18)	25.3 (7.15)	0.08 (218.2)	0.01	−0.25–0.27	0.15
	D (CBG)	11.7 (3.92)	12.2 (4.55)	−0.84 (209.8)	−0.11	−0.36–0.15	0.21
	PA (CBG)	25.3 (5.09)	24.6 (4.70)	1.05 (218)	0.14	−0.12–0.39	0.25
	AB	21.2 (6.93)	21.2 (7.26)	0.02 (217.4)	0.003	−0.25–0.26	0.15
	DE	35.7 (9.00)	35.2 (9.47)	0.53 (215.7)	0.17	−0.19–0.33	0.17
	VI	24.6 (7.68)	25.3 (7.95)	−0.64 (216.1)	−0.09	−0.34–0.17	0.18
	FCV	14.9 (5.26)	14.0 (4.62)	1.32 (221.9)	0.18	−0.09–0.42	0.33
Friend or relative infected by COVID-19							
	EE (MBI)	14.3 (8.80)	13.4 (6.69)	0.36 (14)	0.13	−0.41–0.57	0.29
	D (MBI)	9.4 (8.04)	6.0 (5.09)	1.56 (13.7)	0.43	−0.14–0.88	0.75
	PA (MBI)	21.9 (6.83)	23.4 (6.26)	−0.77 (14.5)	−0.21	−0.68–0.31	0.35
	EE (CBG)	25.0 (6.98)	25.4 (7.17)	−0.19 (14.1)	−0.05	−0.53–0.45	0.28
	D (CBG)	13.4 (5.35)	11.8 (4.15)	1.04 (12.9)	0.30	−0.26–0.77	0.44
	PA (CBG)	23.7 (5.03)	25.0 (4.90)	−0.94 (14.7)	−0.26	−0.72–0.28	0.40
	AB	19.3 (7.93)	21.4 (7.01)	−0.96 (14.4)	−0.26	−0.72–0.27	0.40
	DE	30.3 (12.45)	35.9 (8.89)	−1.66 (13.9)	−0.46	−0.91–0.11	0.86
	VI	24.8 (8.32)	24.9 (7.79)	−0.05 (14.6)	−0.01	−0.50–0.48	0.28
	FCV	16.4 (6.45)	14.3 (4.86)	1.15 (14)	0.31	−0.23–0.77	0.47
Friend or relative died from COVID-19							
	EE (MBI)	13.4 (6.53)	13.7 (8.18)	−0.17 (47)	−0.03	25.4 (7.17)	0.19
	D (MBI)	6.2 (5.31)	6.0 (5.68)	0.17 (51.2)	0.03	−0.30–0.36	0.19
	PA (MBI)	23.2 (6.45)	23.7 (5.45)	−0.53 (57.9)	−0.10	−0.42–0.25	0.22
	EE (CBG)	25.5 (6.98)	24.4 (7.95)	0.81 (49.5)	0.14	−0.20–0.47	0.26
	D (CBG)	11.9 (4.13)	11.7 (4.79)	0.28 (47.4)	0.05	−0.29–0.38	0.20
	PA (CBG)	25.0 (4.72)	24.5 (5.79)	0.56 (47.8)	0.10	−0.24–0.42	0.22
	AB	21.2 (7.09)	21.5 (7.06)	−0.21 (51.5)	−0.03	−0.37–0.30	0.20
	DE	35.5 (9.29)	35.8 (8.98)	−0.22 (54.6)	−0.04	−0.37–0.30	0.19
	VI	24.7 (7.86)	25.8 (7.55)	−0.77 (54.9)	−0.14	−0.46–0.21	0.25
	FCV	14.3 (5.09)	15.3 (4.32)	−1.34 (60)	−0.23	−0.12–0.55	0.43
Psychological effects							
	EE (MBI)	10.8 (6.17)	14.2 (6.82)	−3.36 *** (86.3)	−0.54	−0.82–−0.19	29.22
	D (MBI)	4.7 (4.16)	6.6 (5.6)	−2.64 ** (105.8)	−0.42	−0.71–−0.09	4.18
	PA (MBI)	24.8 (5.62)	22.9 (6.42)	2.07 * (89.2)	0.33	0.003–0.62	1.23
	EE (CBG)	22.4 (7.77)	26.2 (6.77)	−3.01 ** (68.1)	−0.49	−0.78–−0.14	10.80
	D (CBG)	12.2 (3.46)	11.8 (4.44)	0.61 (97.6)	0.10	−0.21–0.40	0.21
	PA (CBG)	26.7 (4.19)	24.5 (5.00)	3.06 ** (87.9)	0.50	0.15–0.79	12.37
	AB	22.0 (5.83)	21.0 (7.38)	1.02 (92.5)	0.17	−0.15–0.46	0.28
	DE	37.7 (7.98)	34.9 (9.47)	2.09 * (89.9)	0.34	0.01–0.63	1.29
	VI	25.9 (7.8)	24.6 (7.8)	1.00 (75.1)	0.16	−0.16–0.46	0.28
	FCV	11.8 (4.33)	15.2 (4.9)	−4.71 *** (88.4)	−0.76	−1.04–−0.40	3548.1
Physical effects							
	EE (MBI)	12.7 (6.23)	14.0 (7.20)	−1.47 (213.4)	−0.20	−0.45–0.71	2.45
	D (MBI)	5.4 (4.82)	6.7 (5.69)	−1.75 (211)	−0.24	−0.49–0.04	0.65
	PA (MBI)	23.7 (6.13)	23.0 (6.43)	0.90 (203.9)	0.12	−0.14–0.37	0.22
	EE (CBG)	23.7 (6.74)	26.5 (7.24)	−2.94 ** (202)	−0.40	−0.65–−0.12	8.22
	D (CBG)	12.3 (3.94)	11.6 (4.43)	1.23 (208.1)	0.17	−0.10–0.42	0.30
	PA (CBG)	25.5 (5.07)	24.6 (4.80)	1.38 (184.8)	0.19	−0.08–0.44	0.37
	AB	21.0 (6.52)	21.4 (7.48)	−0.48 (208.8)	−0.07	−0.32–0.20	0.17
	DE	35.6 (9.59)	35.4 (8.98)	0.11 (188.2)	0.02	−0.24–0.27	0.15
	VI	25.8 (7.31)	24.3 (8.13)	1.44 (207.3)	0.20	−0.08–0.45	0.39
	FCV	12.9 (4.62)	15.6 (4.93)	−4.16 *** (205.6)	−0.56	−0.81–−0.27	407.1

EE (MBI) = Emotional exhaustion, measured by MBI; D (MBI) = depersonalisation, measured by MBI; PA (MBI) = personal accomplishment, measured by MBI; EE (CBG) = emotional exhaustion, measured by CBG-US; D (CBG) = depersonalisation, measured by CBG-US; PA (CBG) = personal accomplishment, measured by CBG-US; AB = absorption; DE = dedication; VI = vigour; FCV = fear of coronavirus. * *p* < 0.05. ** *p* < 0.01. *** *p* < 0.001.

## Data Availability

Not applicable.

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
