# Peer review of "Levels of Burnout and Engagement after COVID-19 among Psychology and Nursing Students in Spain: A Cohort Study"

_ijerph, 2022, doi:10.3390/ijerph20010377_

Round 1
Reviewer 1 Report
Dear Authors, I value the opportunity to provide feedback on the manuscript. Please see below for my assessment.
The title appears “incomplete”; engagement of what?
“Levels of Burnout and Engagement after COVID-19 among Psychology and Nursing Students in Spain: a Cohort Study”
Line 33: “covid pandemic” or “COVID-19 pandemic”?
Abstract: It’s difficult to identify what part is your results? Can you add some numbers please?
In the Introduction, can you add the definition of “burnout” (describe)
In the Introduction, can you add, why did you choose psychology and nursing students? Here, you need to elaborate previous studies that investigated these population. Also, please highlight, what makes these students (psychology and nursing) or populations so important to investigate?
Line 97: Can you clarify; you investigate student population. However, the age range is up to 64 years old? But the mean age is 21?
METHODS
Can you add information regarding reliability of the questionnaires used?
Also, do you have test-retest values for your sample?
Please detail the processes used to cleanse the data in the Methods section. It is important to filter data in order to identify attention patterns, click-through, keystroke analysis, machine replies, etc.
Discussion
Lines 236-248: Can you please consider moving to Methods some of the information here? Please start the Discussion with major/brief findings of your study.
Within the Discussion, can you please indicate, specifically, where the readers should find the information relevant to the contents being discussed? Consider adding Table reference (e.g., Table 1).
Lines 306-307: “On the other hand” seems repeated (consider replacing one of those)
Line 316 and onwards: You need to be more explicit with the strengths and limitations of the study. Need to state clearly.
Line 326: Remove reference from your Conclusion. This section should be the summation of most important points made in Discussion.
Author Response
Thank you very much for your comments and suggestions. In the attached document you can find the answer to each comment.

Reviewer 2 Report
- The manuscript's topic is fascinating and with a growing field of application.
- Author should polish the titles at all levels of the paper, highlight the hierarchy of the paper, and correspond to the technical route of the introduction.
- Paper is well written. Authors should add a little background of the study and limitations of the existing works and clearly explain the contributions at the end of the introduction.
- The motivation of the paper is missing in the Introduction of the paper. Recommended to provide the motivation and need of this work in current scenarios.
- The contributions of the paper may be highlighted in the Introduction.
- What are the limitations of this study?
- The performance comparison should be more comprehensive.
- To broaden the scope of this paper, the authors should refer to some research such as
a- Exploratory data analysis, classification, comparative analysis, case severity detection, and internet of things in COVID-19 telemonitoring for smart hospitals
b- A survey on COVID-19 impact in the healthcare domain: worldwide market implementation, applications, security and privacy issues, challenges and future prospects
- In Conclusion, future directions and challenges should be explained more.
Author Response
Thank you very much for your comments and suggestion. In the attached document you can find the answer to each comment.

Reviewer 3 Report
Interesting, well-written study on the consequences of covid in an academic context.
Recruitment: More details needed describing recruitment and the sample. No detail is given on the university breakdown, or on dates of interview, or how were the students were recruited. A 100% response rate is given but we are left in the dark of how were the students approached, by whom, whether their professors were present while filling up the questionnaires … Was exactly the same questionnaire administered in all university and pre-post-testing… It would also help to know the original purpose of the pre-test survey since it was carried out before covid.
In particular, it is claimed that the pretest and posttest are comparable on a number of dimensions. How about the university/degree composition? You should provide a table showing balance of the pre-post-test samples. There could be differences in the composition of the two samples in this dimension that could partly explain the results.
Since different faculties and degrees were using different strategies, particularly in 2020-21, it would be important to control for degree/university. You could have carried out the equivalence analysis also on this dimension at least or control in a model.
Terminology: In several instances the authors talk of “levels of burnout syndrome”. Should not we talk about “level of burnout” and talk about a syndrome only when the scale is beyond a given level? Also, only t-tests and correlations are provided but not the levels of the variables in the pre- post-test context. That would be important since interpretation should be very different according to whether the differences happen with low or high levels of the variables.
Table 4 is also not clear. In particular, the respective Ns should be provided for easier interpretation. It seems that the lack of effects might be explainable by low statistical power given large intervals in most variables. The exception, psychological effects, could be argued to be measuring something similar. It is notable, in any case, that none of the personal/ friends covid incidence variables are significantly associated with the different scales.
Author Response

(The authors gave the same response as above.)

Reviewer 4 Report
The reviewed work is extremely up-to-date, unfortunately it concerns only selected students from Spanish universities. The problem is much wider, and many students are struggling with emotional problems due to the COVID-19 pandemic. The following shortcomings should be corrected in the work:
1. In the title of Table 1 (line 171 and earlier line 159) it says that it contains data for all variables, does it? This table does not include FCV, gender, marital status, employment situation, etc. The title should be corrected, either statistics for missing variables should be added.
2. A similar remark applies to Table 2.
3. Line 179 says N=190 when it should be 226.
4. Column for FCV are missing in Table 2 (correlations and omega are given only for first 10 variables).
5. Why were the data from both tables not aggregated? Are the distributions of variables significantly different in both populations?
6. In line 161 there is a reference to the omega factor. There is no reference to the source (McDonald, 1999?) and information about the type of indicator or its definition.
7. In line 168, are the statistics for age given in parentheses correctly described?
8. It should be described at least once (before line 187) the column headings of tables 3 and 4 (number of degrees of freedom, difference, etc.).
Author Response

(The authors gave the same response as above.)

Reviewer 5 Report
This is a well written paper that draws on large data for analysis. Personally, I would have liked greater qualitative data to balance the statistical data to illustrate the human elements related to COVID 19 impact on participants. Also, some hint to paradigm shift, policy development going forward should have been framed to develop local and global awareness- after all, the WORLD, was affected by the pandemic, so a global solution needed modelling and this study could have made stronger case here.
Author Response
Thank you very much for your comment. In the attached document you can find the answer to the comment.

Round 2
Reviewer 1 Report
The manuscript has been revised accordingly.
Reviewer 3 Report
The previous comments have been adequately addressed.